# Systematic Observation of the Verbal Behavior of Families of Youth Athletes in Grassroots and Team Sports

**DOI:** 10.3390/ijerph17041286

**Published:** 2020-02-17

**Authors:** Elisa I. Sánchez-Romero, Francisco J. Ponseti Verdaguer, Pere A. Borràs, Alejandro García-Mas

**Affiliations:** 1Department of Social, Legal and Business Sciences, Catholic University of Murcia (UCAM), 30107 Murcia, Spain; 2Department of Pedagogy, University of the Balearic Islands, 07122 Palma, Spain; xponseti@uib.es (F.J.P.V.); pa-borras@uib.es (P.A.B.); 3Department of Psychology, University of the Balearic Islands, 07122 Palma, Spain; alex.garcia@uib.es

**Keywords:** physical activity, grassroots sports, clubs sports, school sport, families, spectators, verbal behavior, competitive environment

## Abstract

Some of the more protective and favorable factors for the development and health in children and teenagers are family and sport, so family involvement in the children’s sports activities is vital in their sports process. The purpose of this study was to analyze the verbal behavior (positive, negative, and neutral comments) of family spectators of school-age athletes regarding sociodemographic and sporting variables. The sample consisted of 190 family spectators of 215 male and female (*M_age_* = 11.66; *SD* = 1.60) football, basketball, and volleyball players. The Parents’ Observation Instrument at Sport Events (POISE) was used for the observation and LINCE was used to codify the verbal comments made. After registering 38,829 comments, the results showed statistically significant differences in relation to the comments made and the gender of athletes, geographical area, kind of sport, and the sporting category. The findings highlight that in a competitive environment, the comments made by spectators related to athletes do not seem to be initiators of potentially violent situations but rather are dependent on the atmosphere in question. Further research is required in this area to foster positive conduct relating to grassroots sports.

## 1. Introduction

In recent years, family participation in grassroots sport has increased significantly [1,2], so that the verbal behavior of parents is the main method of family participation in children’s sporting events [3]. In this sense, one of the consequences is the increase of inappropriate comments by family members and spectators [4]. Thus, Walters et al. [5] recorded over ten thousand verbal comments made by coaches in grassroots sports, of which 35.4% were positive, 21.6% negative, and 43% neutral. The media has also recorded an increase in aggressive behavior and/or violence in sport [6,7]. In this regard, several studies have observed parents that incite aggressiveness in the playing field [8], authoritarian parents who severely sanctioned children in the case of not winning [9], and even parents who exercise violent behavior in the sports field [10,11].

The comments made and the behavior and attitudes shown have an influence on the wellbeing and future performance of athletes [12]. Although it has not been possible to establish a cause-and-effect relationship between inappropriate comments and violent behavior in sport, there is no doubt that such inappropriate comments generate an aggressive atmosphere that, on occasions, can lead to violent behavior. Parental and spectator pressure in games and matches may cause athletes to doubt their moral decisions on behavior, in line with the ethical foundations of the sport [13]. Parents are the most relevant socioeducational agents in sport, so the most successful athletes received more support from their parents [14,15]. Positive parental behavior toward the sporting activity of their children has been found to be associated with the attainment of the values of the sport transmitted by parents [16], evidencing that the family participation in the sports practiced by their children is vital in the sporting process of young athletes [17]. Thus, Witt and Dangi [18] suggest undertaking an intervention with parents to help them to be better s9pectators and, therefore, not to negatively affect performance or athlete attitudes. With the aim of optimizing the integral and sport development of young athletes, these kinds of programs are highly beneficial in sport, given that families are key elements in terms of their influence on athletes, both in a personal and sporting sense [19].

Hence, the aim of this study was to analyze the verbal behavior (categorized into three kinds of comments: positive, negative, and neutral) of family spectators of school-age (9–15 years old) sports teams as well as to study potential sociodemographic and sport-related differences (sex of player, geographical area, sports club, sport modality, and age group). According to previous studies [20,21,22,23], the preliminary hypothesis is: (1) the number of negative comments made in matches played in rural areas will be greater than the number recorded in urban areas; (2) in the verbal behavior of family spectators at basketball games, there will be a higher number of comments (positive, negative, and neutral) compared with the other sports analyzed; and, (3) the number of negative comments made by family spectators will rise as the sporting category increases.

## 2. Materials and Methods

### 2.1. Design

This study corresponds to a predictive correlational design. The possible influence of certain demographic and sporting variables in the verbal behavior of the athletes’ relatives who were watching the matches or games of the grassroots sports teams was measured. The study design allows collecting data and describes connections between two or more variables at a specific point in time [24]. Furthermore, these research designs offer efficiency in the collection of extensive data on a particular subject, while obtaining highly realistic content, which is inherently appealing in solving practical problems. As such, this kind of research design is rarely criticized for being artificial [25]. Therefore, non-participant systematic observation was used as a data collection technique [20]. The observation technique is widely used and accepted in studying changeable social problems in a context of spontaneousness or naturalness of the behavior observed [20,26,27,28,29].

### 2.2. Participants

One hundred and ninety spectators (64 male, 33.7%; 126 female, 66.3%) of 215 athletes of both sexes (164 male, 74.3%; 51 female, 23.7%), aged between 9 and 15 years (*M* = 11.66 years old; *SD* = 1.60), participated in this study. All athletes belonged to 11 sports clubs (68.2% in urban areas; 31.8% rural areas), three grassroots sports: football (50.7%), basketball (14%), volleyball (35.3%), established in different categories: Under-11s (34.4%), Under-13s (49.8%), and Under-16s (15.8%).

#### Ethical Considerations

Authorization to conduct the research was granted by the Regional Government of the Balearic Islands (Spain) in its project, Posam Valors a l’esporty. Furthermore, the Department of Physical Education and Sport of the University of the Balearic Islands (UIB) contacted the different corresponding federations to obtain their authorization and that of their clubs.

This study analyzes the verbal behavior of human beings and, as such, it must meet the ethical principles of respect for human dignity, confidentiality, and non-discrimination. A favorable report by the ethics committee of the University of the Balearic Islands (UIB-93CER18) has been obtained regarding the conducting of this study. Therefore, this study was undertaken in accordance with the 1975 Declaration of Helsinki, revised in 2000.

### 2.3. Instruments

To measure the verbal behavior of the participants, two instruments were used: one for the observation and another to code the behavior observed. Furthermore, sociodemographic and sporting data were asked by researchers (geographical area, sex, sports club, the kind of sport, and sporting category). 

To observe the comments made, the Parents’ Observational Instrument at Sport Events (POISE) [30] was used. Designed to register the verbal behavior of spectators at sports events, it comprises four areas of observation: (1) Nature of the comment (positive, neutral, negative) (Table 1); (2) Target of the comment (players, teams, coaches, officials, other parents, children, spectators, individuals); (3) Event unfolding (ball in play-goal-penalty); and (4) Match or game result (win, loss). It includes a categorization of possible kinds of behavior for each area observed (this study focused on area of observation 1: Nature of the verbal comment made (positive, neutral, negative) and area 4: The match or game result, although it was extended to sporting performance measured in the final classification (low, medium, high). This instrument has an inter-observer and intra-observer reliability rate of 92% and 97%, respectively [31].

To code the comments of family spectators, LINCE [32] was used. LINCE is a coding software that provides computerized procedures in observation methods, facilitating the registering of match actions or spectator comments during the visualization of different match recordings on the same screen. LINCE also helps to simultaneously code match actions and comments to verify the quality of observer data and to export the results obtained to other computer programs for additional analysis [33].

Furthermore, all the match recordings were made using a Toshiba Camileo X-200 video camera (Toshiba Europe GmbH, Madrid, Spain).

### 2.4. Procedure

To analyze the verbal comments of family spectators in the stands at matches, 22 observations were carried out through the POISE, which made one recording per match. It took twenty-two hours and twenty minutes (1332 min) to complete the observation from the stands at the matches of different sports: football (U11s, 60 min; U13s, 72 min); basketball (48 min); and volleyball (average of 60 min per match). 

One of the researchers captured all the entries with a video camera. The researcher only registered audio recordings of the comments made by family spectators at matches and games. Furthermore, if any family member watching the match or game had an issue with the presence of the video camera, the researcher reminded them of the main aim of the study. In terms of the recordings, the researcher was careful to be positioned between the fans of both clubs, depending on the team to be recorded. The researcher always arrived 15 min before the start of each match to detect the location of the family spectators of both teams.

### 2.5. Statistical Analysis

All the statistical analyses were conducted using the SPSS program (IBM, SPSS v.22.0, Armonk, NY, USA). After the verbal behavior of family spectators was registered, it was coded and tabulated to represent the nature of the comments made (positive, neutral, negative). To establish the prevalence of the variable under the study, descriptive statistics (average and standard deviation) and the percentage of comments made by family spectators at grassroots sports team matches and games were calculated. Furthermore, Pearson’s Chi-squared (*χ^2^*) distribution was used to compare the division of categorical variables into three sports. The Poisson regression model was used to estimate the comments according to the variables to compare: geographic area, sex of the player, sports club, kind of sport practiced, sporting category, and sport performance (using the Wald Chi-Squared Test or the Wald *χ^2^* test). For all statistical tests, the level adopted for significance was a two-tailed *p* < 0.05. The effects of the variables of the geographical area, sex of the player, sports club, kind of sport practiced, sporting category, and sports performance in the ratio of comments made by family spectators were also analyzed through the Poisson regression model.

## 3. Results

### 3.1. Prevalence of the Comments Made by Family Spectators

The total of all the comments made per sport is set out in Table 2. A total of 38829 were registered in 22 matches or games observed (11 football, 4 basketball, 7 volleyball) at a rate of 29.15 (95% confidence interval (CI): 22.45−37.82) comments per minute (ratio). The highest number of registered comments made by family spectators at matches or games corresponded to football (*n* = 18,024), which also had a higher play-observation time (720 min), followed by volleyball (*n* = 12527; 420 min) and, lastly, basketball games, with the lowest rate (*n* = 8278; 192 min). However, the highest number of comments per minute was registered in basketball matches (43.10; 95% CI: 10.92−75.58), followed by volleyball games (29.83; 95% CI: 16.04−43.67), and lastly by football matches (25.03; 95% CI: 14.28−36.81). That said, no statistically significant differences were observed between the number of comments made per minute (ratio) in the three sports observed (Wald *χ^2^*_(2)_ test (*n* = 22) = 2.42, *p* > 0.05).

In terms of the nature (positive, neutral, negative) of all the comments made, the highest rate corresponded to neutral comments (*n* = 21081; 54.29%), followed by positive comments (*n* = 13053; 33.62%) and, lastly, negative comments (*n* = 4695; 12.09%). After establishing the comments made according to their nature and classifying them by sport, volleyball registered the highest percentage of neutral comments (*n* = 7153; 57.10%) and a lower rate of negative comments (*n* = 1346; 10.74%) compared with the other sports observed. The sport that registered the highest percentage of positive comments was football (*n* = 6427; 35.66%), while basketball registered the highest number of negative comments (*n* = 1194; 14.43%) (Table 3). In this respect, statistically significant differences were detected between the comments made by family spectators (positive, neutral, negative), according to the sport of the matches or games observed (Pearson’s *χ^2^*_(4)_ test (*n* = 38829) = 44.00, *p* < 0.001; Cramer’s *V* = 1.00, *p* < 0.001). 

### 3.2. Effects of the Variables in the Positive Comments Made by Family Spectators

After conducting the Poisson regression analysis (*n* = 22), statistically significant differences were detected in the relationship between the variable of positive comments made by family spectators and that of the kind of sport practiced (Wald *χ^2^*_(2)_ test = 6.50, *p* < 0.05).

With regard to the kind of sport practiced (football, basketball, or volleyball), using football as the point of reference, the incidence rate [Exp (*β*)] of positive comments was 1.51 (95% CI: 1.09–2.10) for basketball. That entails that when basketball games were played, the rate of positive comments made by family spectators increased 51% on average compared with football matches.

However, no statistical differences were detected in relation to the variable of positive comments made by family spectators and the geographical area variable (Wald *χ^2^*_(1)_ test = 0.04, *p* > 0.05), the variable of the sex of players (Wald *χ^2^*_(1)_ test = 2.47, *p* > 0.05), the sporting category variable (Wald *χ^2^*_(2)_ test = 6.35, *p* > 0.05), and the sports club variable (Wald *χ^2^*_(10)_ test = 24.27, *p* > 0.05).

### 3.3. Effects of the Variables in the Neutral Comments Made by Family Spectators

The Poisson regression analysis (*n* = 22) produced statistically significant differences in the relationship between the variable of neutral comments made by family spectators and: (1) the variable of sex of the players (Wald *χ^2^*_(1)_ test = 8.05, *p* < 0.05); (2) the sporting category variable (Wald *χ^2^*_(2)_ test = 7.80, *p* < 0.05); and (3) the sport practiced variable (Wald *χ^2^*_(2)_ test = 8.04, *p* < 0.05).

With regard to the sex of players, using the male sex as a reference, the incidence rate [Exp (*β*)] of neutral comments was 1.89 (95% CI: 1.22–2.93) in female matches and games. That means that when female teams played, the rate of neutral comments made by family spectators increased 89% on average compared with football matches played by male teams.

In terms of sporting category (U11s, U13s, and U16s), using the U11s as a reference category, the incidence rate [Exp (*β*)] of neutral comments was 0.18 (95% CI: 0.04–0.77) for the U16s. That means that in U16 matches or games, the rate of neutral comments made by family spectators fell 82% on average compared with those of the U11s.

With regard to the kind of sport practiced (football, basketball, or volleyball), using football as the reference category, the incidence rate [Exp (*β*)] of positive comments was 1.96 (95% CI: 1.18–3.28) for basketball. That means that when basketball games were played, the rate of neutral comments made by family spectators increased 96% on average compared with football matches.

No statistically significant differences were detected in the relationship between the neutral comments of family spectators and the geographical area variable (Wald *χ^2^*_(1)_ test = 0.05, *p* > 0.05) and the sports club variable (Wald *χ^2^*_(10)_ test = 23.34, *p* > 0.05).

### 3.4. Effects of the Variables in the Negative Comments Made by Family Spectators

After undertaking the Poisson regression analysis (*n* = 22), statistically significant differences were detected in the relationship between the variable of negative comments made by family spectators and: (1) the geographical area variable (Wald *χ^2^*_(1)_ test = 15.62, *p* < 0.001); (2) the sporting category variable (Wald *χ^2^*_(2)_ test = 15.75, *p* < 0.001); and (3) the sport practiced variable (Wald *χ^2^*_(2)_ test = 18.38, *p* < 0.001).

In terms of the geographical area (urban and rural areas), using the urban areas the reference category, the incidence rate [Exp (*β*)] of negative comments was 1.53 (95% CI: 1.24–1.89) in games or matches played in urban area. That means that when the games or matches were played in rural areas, the rate of negative comments made by family spectators increased 53% on average compared with the games or matches in urban areas.

In terms of the sporting category (U11s, U13s, and U16s), using the U11s as a reference, the incidence rate [Exp (*β*)] of negative comments was 1.28 (95% CI: 1.02–1.66) for the U13s. In this regard, when U13 games or matches were played, the rate of negative comments made by family spectators increased 28% on average compared with U11 games or matches.

With regard to the kind of sport practiced (football, basketball, or volleyball), using football as the reference category, the incidence rate [Exp (*β*)] of negative comments was 1.76 (95% CI: 1.40–2.29) for basketball. As such, when basketball games were played, the rate of negative comments made by family spectators increased by 76% on average compared with football matches.

However, no statistically significant differences were detected in relation to the variable of negative comments of family spectators, the variable of sex of the players (Wald *χ^2^*_(1)_ test = 2.51, *p* > 0.05), and the sports club variable (Wald *χ^2^*_(10)_ test = 75.81, *p* > 0.05).

## 4. Discussion

The aim of this study was to analyze the verbal behavior (categorized into three kinds of comments: positive, negative, and neutral) of family spectators of school-age (9–15 years old) sports teams as well as to study potential sociodemographic and sport-related differences (sex of player, geographical area, sports club, sport modality, and age group).

The results obtained allow us to ensure that hypothesis 1 has been confirmed. In effect, there are statistically significant differences in the negative comments made, according to the geographical area variable, with a 53% increase in games and matches in rural areas. In this sense, it coincides with previous authors [20], who have concluded that the sociocultural context is an important variable between the behavior of spectators at sporting events and how sporting success is interpreted. The reason is due to the urban areas are characterized by having a higher population density and human diversity. Conversely, in rural areas availability, people manage to create and develop a sense of belonging to the territory. Moreover, results have shown that, according to previous studies [20], some clear connections between the comments made by family spectators and the sport atmosphere.

Furthermore, the second hypothesis has been confirmed, because in basketball the most of comments were negative, in contrast with previous studies about other psychosocial factors [21,22]. Moreover, a higher percentage of comments was observed (51% positive; 96% neutral; and 76% negative) compared with the other sports, showing statistically significant differences. Basketball is a sport in which there is constant physical contact between players, as well as blocks, and covering with and without the ball, which leads to numerous heated moments. Furthermore, spectators are very close to the court, which, together, could generate more emotion in the stands, producing a higher number of comments as a result. It is evident, however, that clarification is required on this shift through further studies, as it is still to be “observed” by the general media. Negative comments from parents cause more pressure, insecurity, anxiety, and feelings of guilt, leading to a reduced sporting performance [20], while also may generate negative psychosocial effects in their kids [10].

The third hypothesis was also confirmed, given that statistically significant differences were identified in the negative comments made according to the sporting category (28% more were registered in the U13s than in the U11s category). In contrast with our findings, in recent studies [23,34], no association between the age of athletes and the verbal aggression of spectators was found, related to some theories about [35].

Finally, our results have shown that the comments with the greatest prevalence during games and matches were the neutral ones, followed by positive and, lastly, negative comments, which coincides with Reference [5]. In terms of the kind of sport, the highest percentage of positive comments was in football, while basketball had the highest number of negative comments, contrary to general belief. This “shift” has already been observed in antisocial situations, such as in the willingness to accept gamesmanship and cheating [22]. In terms of the number of comments made per minute (ratio), the highest number was in the basketball games, followed by volleyball games and, lastly, in the football matches. This distribution is fairly similar, excluding the theoretical distances, to the average observations made with the CBAS (Cognitive-Behavioral Approach Skills) regarding coaches and the kind of instructions they give to their players [21].

### Limitations and Future Developments

The verbal behavior of family members in the stands is an important form of communication, as it represents their main method of participation in their children’s sporting events [3]. These same authors suggest that the description of these kinds of behavior in games and matches could be an aspect of improving knowledge of parent–child relations, as well as the rules and expectations experienced by parents in organized youth sports.

It is important, therefore, to conduct research focused on analyzing not only comments but also the behavior of family members, spectators, and sport-related agents at school-age sports games and matches. The use of non-participant systematic observation in the research provides an approach to social problems in a natural context of observed behavior [20]. To do that, studies based on observation and description of this kind of aspect, as considered in this research, would serve as a starting point from which to discover the current situation regarding the atmosphere that surrounds athletes on the field of play. This kind of studies could also lead to the implementation and development of observation and registration protocols, as well as to their validation, in multiple areas of sport and physical activity, such as fair play in youth football [36], the influence of contextual variables in the efficiency of handball goalkeepers [37], and sports leadership behavior in youth football coaches [22,38].

Furthermore, as suggested by Walters et al. [5], the rate of negative comments observed in all sports is cause for concern, particularly considering the young ages of the child athletes. Even though, in this study, the percentage of negative comments (12.09%) was below those of neutral and positive comments, fostering positive behavior (verbal and non-verbal) in everything that surrounds the physical activity and sport of children is important.

Similarly, analyzing comments according to the sex of family members is an interesting line to consider for future research. However, due to the number of comments registered in this study (*n* = 38,829), coding this variable was too difficult or challenging. As such, there is the possibility of conducting future observations based on verbal comments (and other variables and aims geared toward their children’s sport) of a smaller number of parents, previously selected, and carried out with other data collection techniques, such as semi-structured interviews, like those conducted in Reference [3]. The findings show that when female teams played, the rate of neutral comments made by family spectators significantly increased by 89% on average compared with matches played by male teams. The question of gender, therefore, is a variable to study, due not only to the comments received by female athletes but also to the comments made by male and female in the stands. However, the two future studies that we believe most promising that can be developed in the short term are: 1) take into account the changes in the score of the matches (raw performance) to check if there is a correlation between the result and the typology of the comments of the parents/spectators at any given time [3]; and 2), since it has been proven that the collective efficacy can predict the performance of a team -in semi-professional football players, it would be very interesting to study whether individual and collective effectiveness is modified according to the nature of the comments made from the sideline, considering that it can be understood as one of the basic sources of self-efficacy (verbal persuasion).

## 5. Conclusions

The main conclusion that can be taken from this study, conducted with a rather complex method of observation in real competition situations, is that the comments made by family spectators do not appear to be “initiators” of potentially violent on-pitch or court situations, but rather they depend on the atmosphere created, which can include group pressure, cognitive dissonance, and involvement. Therefore, from an applied perspective, psychoeducational endeavors, until now exclusively aimed at coaches and parents, should be geared toward the sporting culture and structures of clubs. Club sports participation may be an important component in the promotion of physical fitness and healthy at younger ages [39]. The way in which spectators create their reference framework for interpreting sporting success is mediated by sociocultural context, regarding which the media has an important role to play [25].

## Figures and Tables

**Table 1 ijerph-17-01286-t001:** Categories of the kind of comments made by family spectators [30].

Positive	Neutral	Negative
- Reinforcing: comments aimed at reinforcing and supporting the behavior of athletes (e.g., “well done”). - Hustle: done with the aim of encouraging athletes so that they improve performance (e.g., “go on, go on, go on”).	- Instructing: telling players what to do (e.g., “play up the field/court”). - Direct question (e.g., “Do you want to come off?”). - Indirect question: aimed at a player, but not relating to the event (e.g., “Who will be at training next week?”). - Rhetorical question: one that does not require a response (e.g., “Where was the movement today?”). - Social: any comment not related to the event (e.g., “Let’s get a coffee later”). - Other: any other comment that does not fit into another category.	- Correction: comments changing specific behavior. The comment is usually directly related to the subject (e.g., “John, arms up”). - A telling off: a comment indicating that the performance was not good enough. Comments displeasure with the circumstance (e.g., “Don’t sit there, get up”). - Witticism: a comment that often involves sarcasm or ridicule (e.g., “Your dad could hit it better than that”). - Contradicting: comments that could vary and that players could find confusing (“hit the ball toward the center. You should have hit it to the right-hand side!”).

**Table 2 ijerph-17-01286-t002:** Games and matches, comments, minutes, and ratio of comments observed per minute.

Sport	Matches/Games	Comments	Minutes	Ratio * (95% CI)
Football	11	18024	720	25.03 (14.28−36.81)
Basketball	4	8278	192	43.10 (10.92−75.58)
Volleyball	7	12527	420	29.83 (16.04−43.67)
Total	22	38829	1332	29.15 (22.45−37.82)

* Comments per minute.

**Table 3 ijerph-17-01286-t003:** Average, standard deviation, number, and percentage of comments (according to their nature and established by sport).

Comments							
	Positive		Neutral		Negative		Total
Sport	*M ± SD*	*n* (%)	*M ± SD*	*n* (%)	*M ± SD*	*n* (%)	*n* (%)
Football	9.09 ± 4.23	6427 (35.66%)	3.18 ± 4.07	9442 (52.39%)	13.18 ± 10.90	2155 (11.96%)	18024
Basketball	13.75 ± 6.13	2598 (31.39%)	6.25 ± 6.19	4486 (54.20%)	23.25 ± 9.18	1194 (14.43%)	8278
Volleyball	9.57 ± 5.06	4028 (32.15%)	3.14 ± 2.61	7153 (57.10%)	17.14 ± 9.19	1346 (10.74%)	12527
Total	10.09 ± 4.94	13053 (33.62%)	3.73 ± 4.10	21081 (54.29%)	16.27 ± 10.36	4695 (12.09%)	38829

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
