# Peer review of "Systematic Observation of the Verbal Behavior of Families of Youth Athletes in Grassroots and Team Sports"

_ijerph, 2020, doi:10.3390/ijerph17041286_

Round 1

Reviewer 1 Report

General comments

Firstly, I would like to thank the editor for giving me the opportunity to review the manuscript titled “Systematic observation of the verbal behavior of families of youth athletes in grassroots sports”. I have to admit that I enjoyed a lot reading this manuscript, mainly (but not solely) because the topic addressed is very interesting and trendy nowadays. In general speaking terms, the design and methodology used in this manuscript are robust and widely supported by previous works. The results are well written and the practical applications that can merge are very promising in term of designing tailored intervention programs. However, I think authors should address few aspects that I have highlighted below before considering this manuscript as accepted for publication is this very repute journal.

Specific comments

INTRODUCTION

The introduction section is too long (66 lines and approximately 960 words) and does not flow well. The authors should have focused this section on the problem object of study rather than on other aspects that were not covered or that were too general (please be as much specific as possible). Thus, I suggest that the introduction section should:

Describe the relevance / impact (magnitude of the problem) of the topic selected (i.e. the description of the verbal behavior of families of youth athletes in grassroots sports as well as the study of the potential sociodemographic and sport-related differences) on the young athlete development (i.e. possible development of violent conducts during match play, likelihood of abandoning / giving up the sport practice, impact on the acute and chronic sport performance, psychological characteristic related to sport performance [stress, motivation, etc.] …). Briefly analyze the findings reported by previous studies (i.e. what is already know) (e.g.: has any study documented / reported sociodemographic and sex-related differences?). Identify the main limitations of the scientific literature (please guide this point toward what limitation the current study wants to address). Clearly specify the importance of addressing the limitation that the current study has selected (rationale of the study) and the potential practical applications that may be generated (please make clear that the results of this study could help to design tailored intervention programs to reduce the magnitude / frequency of the aspects mentioned in the point a).

All these points should be addressed in no more than 3-4 paragraphs and 750 words.

The objective of this study perhaps may be re-formulated to be better adapted to the topic covered. I suggest:

To analyze the verbal behavior (categorized into three kinds of comments: positive, negative and neutral) of family spectators of school-age (9-15 years old) sports teams as well as to study potential sociodemographic and sport-related differences (sex of player, geographical area, sports club, sport modality and age group).

When possible, the hypotheses of any study should be formulated considering the available scientific evidence. Thus, I suggest that the authors indicate in the text whether or not they formulated their initial hypothesis taking into account the findings showed in previous studies.

Minor comments:

The expression “among other aspects” has been used several times.

MATERIALS AND METHODS

This section is well written.

The authors have justified with merit the instruments selected to record de data and the procedure carried out has been described in detail, which would allow that other researchers can replicate this study.

Perhaps, instead of speaking about “Palma” and the "areas outside of Palma", I am wondering if authors could use the terms “rural and urban areas”. If this distinction were not possible, it would be great to know what aspects describe Palma and what are the main differences between Palma (as geographical area) and the rest of the areas that have been included in this study

In order to improve the validity of the statistical analysis conducted, I suggest that the authors used a Bayesian approach instead of the tradition frequentist techniques they have used to analyze the data. Some good reasons that support this comment can be found in the following articles:

Halsey, L. G. (2019). The reign of the p-value is over: what alternative analyses could we employ to fill the power vacuum?. Biology letters15(5), 20190174 Amrhein, V., Greenland, S., & McShane, B. (2019). Scientists rise up against statistical significance. Nature, 567, 305–307.

This new statistical approach quantifies the degree of evidence to support or refuse the null or alternative hypothesis through the interpretation of the Bayesian factor. For more information, please check:

Wagenmakers, E. J., Love, J., Marsman, M., Jamil, T., Ly, A., Verhagen, J., ... & Meerhoff, F. (2018). Bayesian inference for psychology. Part II: Example applications with JASP. Psychonomic bulletin & review25(1), 58-76. Vandekerckhove, J., Rouder, J. N., & Kruschke, J. K. (2018). Bayesian methods for advancing psychological science.

RESULTS

This section is also very well written, nothing to say with the exception that it is likely that the new statistical approach I have just suggested may change some results.

DISCUSSION

Probably, this is the weakest section of this manuscript. It should be re-written. Below I suggest some aspects that may improve the reading of this section.

This section should be focused on discussing the results found in this study. That is to say, the authors should not only summarize the main results but also explain the reason/s behind them. Afterward, the authors should compare their results with the findings showed in previous studies and when necessary, analyze the inter-studies differences and suggest some possible reasons that may justify these differences.

Thus, and for example, in lines 257-265 were described the differences observed between the comments registered during games and matches outside of Palma and those played in it. However, the authors did not mention any reason that may explain these differences. Furthermore, these results were not compared with the ones found by previous studies.

Similarly, in lines 266-275, although the authors described and justified the reason for having found sport-related differences they did not compare these results with the ones reported by other previous studies.

Lines 276-280, Why have the number of negative comments from parents moved up through the age groups? Why your results are different to the ones reported by Wann & Sherman (2019).

LIMITATIONS

You should describe here the limitations concerning to your study and not the ones present in the scientific literature.

FUTURE RESERACH LINES

Please, and again, in this section you should focus everything you write on the topic you addressed in this study and after considering your results as starting point, please suggest some new studies / research lines that may cover the limitations and / or aspects that you could not address.

Reviewer 2 Report

I congratulate the authors for conducting such an interesting study. The present study aimed to investigate verbal behaviour among the family spectators of the school-age athletes. However, numerous problems regarding the research were observed. For example, it seems that the amount of explanations within the text is unbalanced. I just add some minor comments, and hopefully, it will be published soon.

The manuscript was written in sloppy English, making it hard to comprehend. Please send this manuscript for English proofreading. Title: since it is team sports, please add “Team”. Pg1, Line 19 – LINCE (full name) The introduction needs more justification for the purpose of this study. The literature review was not sufficient linking to the purpose of the study. Please include sample size calculation. Please justify the purpose of categorizing the groups into Under-11, Under-13, and Under-16. Any difference? I felt the study need stronger justification, stronger conceptual framework and need

Unfortunately, I feel that the current manuscript needs more attention to the required level for publication.

Thank you.

Reviewer 3 Report

It's important as well as interesting to explore the potential influence of the verbal behaviors of family spectators on youth athletes. I believe the literature review provides the audience with a good general background of such research. However, I think a more comprehensive literature review is needed for the proposed hypotheses. For example, the authors proposed that "the  number of negative comments made in matches or games played outside of Palma will be greater than the number recorded in Palma." However, limited research was cited in the literature review to support this hypothesis. In other words, I feel the hypotheses proposed in the study is lack of support from previous research. In addition, I think additional efforts is needed for the edits of the writing of literature review. For example, in stead of using the citation numbers, the authors may want to use the researchers' names while citing them to reduce misunderstanding.

Methodologically, although the authors call the method of this study a "observation", I think it might be more accurate to classify it into "content analysis". The authors video taped the verbal behaviors of parents, and then conducted the content analysis of these video-taped comments. I suggest the authors to include the procedure of their content analysis into the method section. For example, how did they develop the coding book? What's the criteria for coding the comments into "positive" or "negative" categories? How many coders were involved in the coding procedure? How did they develop the inter-coder reliability? Was there any measurements used to test the inter-coder reliability of this study? and so on.

I wish the authors luck with the work on improving the quality of this manuscript.    

Round 2

Reviewer 1 Report

Dear editor and authors

The authors have addressed all the suggestion I made in my first review and hence, from my point of view, the manuscript has been improved.

However, before considering this manuscript acceptable for publication I would like that the authors correct some minor mistakes that I have written below:

Line 32, the words “the verbal behavior of parent” have been written twice, please remove one.

Line 35, after “Thus,” the name of the authors that corresponding with the reference number 5 is missed, is this a mistake or is the way it should be written? The same circumstance happens again in line 51. Please check the whole manuscript and amend these mistakes when needed

Lines 57-61. Perhaps this item of information should be moved to the method section as it is justifying the observational methodology selected for the authors to address the purpose of this study

Line 253 “recent studies” is written but only a single reference is provided. Please amend it
